

# Droughts in Germany: Performance of Regional Climate Models in Reproducing Observed Characteristics

Dragan Petrovic[1], Benjamin Fersch[1] and Harald Kunstmann[1,2]

[1]Institute of Meteorology and Climate Research (IMK-IFU), Karlsruhe Institute of Technology, Campus Alpin, Kreuzeckbahnstraße 19, 82467 Garmisch-Partenkirchen, Germany
[2]Institute of Geography and Center for Climate Resilience, University of Augsburg, Alter Postweg 118, 86159 Augsburg, Germany

*Correspondence to*: Dragan Petrovic (dragan.petrovic@kit.edu)

**Abstract.** Droughts are among the most relevant natural disasters related to climate change. We evaluated different regional climate model outputs and their ability to reproduce observed drought indices in Germany and the near surroundings between 1980-2009. Both, outputs of an ensemble of six EURO-CORDEX models of 12.5 km grid resolution and outputs from a high resolution (5 km) WRF run were employed. The latter was especially tailored for the study region regarding the physics configuration. We investigated drought related variables and derived the 3 month Standardized Precipitation Evapotranspiration Index (SPEI-3) to account for meteorological droughts. Based on that, we analyzed correlations, the 2003 event, trends and drought characteristics (frequency, duration and severity) and compared the results to E-OBS. Methods used imply Taylor diagrams, the Mann-Kendall trend test and the Spatial Efficiency (SPAEF) metric to account for spatial agreement of patterns. Averaged over the domain, meteorological droughts were found to occur approx. 16 times in the study period with an average duration of 3.1 months and average severity of 1.47 SPEI units. WRF's resolution and setup was shown to be less important for the reproduction of the single drought event and overall drought characteristics. Depending on the specific goals of drought analyses, computation resources could therefore be saved, since a coarser resolution can provide similar results. Benefits of WRF were found in the correlation analysis. Greatest benefits were identified in the trend analysis: Only WRF was able to reproduce the observed negative SPEI trends in a fairly high spatial accuracy, while the other RCMs completely failed in this regard. This was mainly due to the WRF model settings, highlighting the importance of appropriate model configuration tailored to the target region. Our findings are especially relevant in the context of climate change studies, where the appropriate reproduction of trends is of high importance.

## 1 Introduction

In the recent past, Germany and other parts of Central Europe have been hit by dryness in the summer periods. Especially the severe drought events in 2015 (e.g. Hoy et al., 2017; Ionita et al., 2017; Laaha et al., 2017), 2018 (e.g. Bastos et al., 2020; Thompson et al., 2020) and 2019 (e.g. European Drought Observatory, 2019; Boergens et al., 2020; Hari et al., 2020;



Ziernicka-Wojtaszek, 2021), which occurred in combination with heatwaves, have contributed to this. In addition, 2020 was also categorized as too dry, mainly in the spring and summer months (DWD, 2020; Umweltbundesamt, 2021). These events have contributed to increased awareness of climate extreme events in the affected regions.

There are studies that suggest an increasing trend (e.g. Dai, 2011, 2013; Sheffield et al., 2012; Trnka et al., 2016), a decreasing trend (e.g. Spinoni et al., 2014) and no trend (e.g. Spinoni et al., 2019; Oikonomou et al., 2020; Vicente-Serrano
et al., 2021) for droughts for the past decades in the Central European region. The discrepancies in the findings are due to the complex characteristics and several different ways of defining (Mishra and Singh, 2010; Lloyd-Hughes, 2014; Crausbay et al., 2017) and quantifying (Wilhite and Pulwarty, 2007; Vicente-Serrano, 2016) a drought event. Moreover, different analysis periods (Hannaford et al., 2013) and a broad range of usable meteorological variables (Vicente-Serrano et al., 2021) lead to uncertainty in drought trends. Economically, however, there was a clear increase in the costs caused by drought
events in the past in the EU (EEA 2010).

In this study, we conduct a drought analysis for the time period 1980-2009 in Germany and the near surroundings by employing an ensemble of regional climate models (RCMs). We are constrained to that time period because of the data availability in the RCM runs.

For Europe, the availability and reliability of RCM simulations have evolved rapidly in the last years (Štepánek et al., 2016).
Concerted downscaling projects and initiatives like PRUDENCE (Christensen and Christensen, 2007), ENSEMBLES (van der Linden and Mitchell, 2009) and most recent CORDEX (Giorgi et al., 2009) have contributed to this development. Several studies, using drought-related data from CORDEX outputs, have been conducted in the past for different parts of the world, the majority with focus on future development of drought under climate change, some with focus on past events. For the EURO-CORDEX domain, there have been studies dealing with the evaluation of the EURO-CORDEX RCM's capability
in historical drought reproduction in Italy (Peres et al., 2020), the comparison and evaluation of drought indices in Poland (Meresa et al., 2016), the future development of drought conditions under different scenarios for the Czech Republic (Štepánek et al., 2016; Potopová et al., 2018), Romania (Dascălu et al., 2016), Poland (Meresa et al., 2016) and entire Europe (Spinoni et al., 2018). Regarding the rest of the globe, studies have been carried out focusing on the evaluation of the CORDEX RCM's ability in simulating historical droughts and their characteristics over West Africa (Diasso and Abiodun,
2017), East Asia (Um et al., 2017) and Bangladesh (Chowdhury and Jahan, 2018). Furthermore, there have been analyzes of climate change impacts on droughts and their characteristics in the future for the Mediterranean region (Marcos-Garcia et al., 2017), India (Das and Umamahesh, 2018), Iran (Senatore et al., 2019), Vietnam (Nguyen-Ngoc-Bich et al., 2021) as well as for the entire globe (Spinoni et al., 2020). In these studies, different drought indices have been used to identify droughts and describe their characteristics. Among the most common ones are the Standardized Precipitation Index (SPI) and
Standardized Precipitation Evapotranspiration Index (SPEI) (Meresa et al., 2016; Diasso and Abiodudun, 2017; Marcos-Garcia et al., 2017; Um et al., 2017; Das and Umamahesh, 2018; Potopová et al, 2018; Spinoni et al., 2018; Spinoni et al., 2020), the Palmer Drought Severity Index (PDSI) (Dascălu et al., 2016; Chowdhury and Jahan, 2018; Nguyen-Ngoc-Bich et al., 2021) and the self-calibrated PDSI (scPDSI) (Senatore et al., 2019). Additionally, some self-developed or less common





used indices were applied: The Standardized Runoff Index (SRI) (Meresa et al., 2016), the Standardized Flow Index (SFI)
(Marcos-Garcia et al., 2017) and the Reconnaissance Drought Indicator (RDI) (Spinoni et al., 2018).

So far, according to our knowledge, there is no study that presents an evaluation on the capability of EURO-CORDEX
RCMs to reproduce droughts and their characteristics with focus over Germany, which we therefore would like to address in
this study. There is a large number of studies dealing with the performance of RCMs in terms of correct reproduction of
meteorological variables. Emphasis is often on temperature and precipitation and effects of different model resolutions and
physics parameterizations are investigated. There are different findings concerning the effects of increased model resolution
on precipitation, the most important variable for droughts. They strongly depend on the season, precipitation amount and
region. Regarding extreme events and summer precipitation, especially in complex terrain, higher model resolution usually
seems to be beneficial (e.g. Rauscher et al., 2010; Tripathi and Dominguez, 2013; Lee and Hong, 2014; Olsson et al., 2015;
Torma et al., 2015; Prein et al., 2016; Rauscher et al., 2016; Dieng et al., 2017; Vichot-Llano et al., 2021). In terms of winter
precipitation and annual mean patterns, there are often no distinct differences between coarse and fine resolution (e.g.
Rauscher et al., 2010; Tripathi and Dominguez, 2013; Kotlarski et al., 2014; Casanueva et al., 2016; Dieng et al., 2017;
Vichot-Llano et al., 2021). Compared to precipitation, there are less studies examining the effects of increased model
resolution on simulated air temperature, the second most important variable for droughts. Vautard et al. (2013) employed an
ERA-interim driven EURO-CORDEX ensemble of 12.5 and 50 km resolution for heatwave analysis over Europe between
1989 and 2008. Increased resolution was shown to induce temperature 90[th] percentile warming and cooling for some models.
It also led to reduced biases in the heat wave reproduction. Zeng et al. (2016) and Vichot-Llano et al. (2021) found that
temperature fields are better reproduced with higher resolution, while Di Luca et al. (2013) conclude a low potential for
added value of increased resolution. They see the highest added value mostly in regions with important surface forcing like
complex topography or land-water contrasts.

Every model simulation requires a suited setup regarding the domain configuration and physical parameterizations for the
selected target region (e.g. Stoelinga et al. 2003; Kumar et al., 2010). To find appropriate settings, usually the skill of
different parameterizations for temperature and precipitation is evaluated with respect to observations. Vautard et al. (2013)
also analyzed possible sources of model spread. The simulation of hot temperatures was shown to be primarily sensitive to
the convection and microphysics schemes, which has effects on the incoming energy and the Bowen Ratio. They further
found that a large part of the model spread can be attributed to parameterizations and that parameterizations can have
different impacts depending on the spatial resolution. Mooney et al. (2013) tested the effects of 12 combinations of physical
parameterizations in WRF over Europe on surface temperature, precipitation and mean sea level pressure. They utilized two
longwave radiation schemes, two land surface models (LSM), two microphysics schemes and two planetary boundary layer
(PBL) schemes. They found that temperature shows the greatest sensitivity to the LSMs, some sensitivity to the radiation
schemes in winter and little sensitivity to the microphysics and PBL schemes. Precipitation showed sensitivity to the LSM
especially in summer. This is also valid for the radiation and the microphysics schemes, but to a lesser extent. There was
only negligible sensitivity to the PBL schemes. They concluded a strong dependence on region and season of the optimal





parameterization combination. Kotlarski et al. (2014) emphasize the high importance of model configurations by describing that, in case of temperature, the "bias spread across different configurations of one individual model can be of a similar magnitude as the spread across different models".

In this study, we accordingly investigate the effects of increased model resolution and model settings on the reproduction of a drought index and thereby fill another research gap. For this reason, we analyze a variety of RCM simulations, i.e. a 5 km three domain WRF run, and an ensemble of six EURO-CORDEX realizations at 12.5 km horizontal resolution. Ideally, computational resources could be saved, if RCMs with coarser grids were able to yield likewise performance as their better resolved counterparts. The WRF model setup was thoroughly determined for Germany. The physics combinations were chosen so that the combined biases of air temperature and precipitation are as small as possible (Wagner and Kunstmann, 2016; Warscher et al., 2019), while the configurations of the EURO-CORDEX RCMs were setup for the entire EUR-11 domain of CORDEX (Giorgi et al., 2009). Since the WRF run was concerted at the study region and has a higher resolution, one may expect better performance regarding the reproduction of air temperature and precipitation and thus likely also of drought indices compared to the EURO-CORDEX runs. To attribute possible better WRF performances to resolution or setting effects, we are able to use the second domain of the WRF run which has 15 km resolution, hence it is slightly coarser than the EURO-CORDEX simulations. Thus, the main objectives of the study are as follows:

1. To evaluate the performance of regional climate models in reproducing the SPEI drought index and related drought characteristics employing a six-member EURO-CORDEX ensemble and a high resolution WRF run. The EURO-CORDEX RCMs and WRF differ in resolution (12.5 km vs. 5 km), while the model physics configurations differ among every single RCM.

2. To gain insights into drought development for Germany and the near surroundings between 1980-2009.

Therefore, the results are evaluated and compared to observations. Specifically, we analyze precipitation and temperature reproduction, SPEI correlations and trends, related drought characteristics and additionally the drought event 2003. The characteristics include frequency, duration and severity and are based on SPEI time series.

## 2 Data

### 2.1 EURO-CORDEX model simulation data

We employed an ensemble of six EURO-CORDEX RCM simulations. The experiments were performed with 0.11° (≈ 12.5 km) horizontal grid resolution, covering the EUR-11 CORDEX-Domain. Data from the following RCMs was used: COSMO-CLM, ALADIN 6.3 (hereafter referred to as ALADIN in the text), REMO2015 (REMO), RegCM 4.6 (RegCM), RACMO 2.2e (RACMO) and RCA4 (see Table 1 for more information). These model simulation outputs were selected due to their availability and because they cover the study period 1980-2009. All runs obtained their boundary conditions from the global ERA-Interim reanalysis (Dee et al, 2011).





**Table 1.** Overview of the EURO-CORDEX RCMs used in this study.

| Experiment ID | Institution | RCM name | RCM description |
|---|---|---|---|
| CLMcom_ETH-COSMO-crCLIM-v1-1 | Eidgenössische Technische Hochschule Zürich (ETH) Zürich in collaboration with the Climate Limited-area Modeling (CLM) Community | COSMO-CLM | Rockel et al., 2008 |
| CNRM-ALADIN63 | Centre National de Recherches Meteorologiques (CNRM) | ALADIN 6.3 | Daniel et al., 2019 |
| GERICS-REMO2015 | Helmholtz-Zentrum Geesthacht, Climate Service Center Germany (GERICS) | REMO2015 | Pietikäinen et al., 2018 |
| ICTP-RegCM4-6 | International Centre for Theoretical Physics (ICTP) | RegCM 4.6 | Giorgi et al., 2012 |
| KNMI-RACMO22E | Royal Netherlands Meteorological Institute (KNMI) | RACMO 2.2e | Van Meijgaard et al., 2012 |
| SMHI-RCA4 | Swedish Meteorological and Hydrological Institute, Rossby Centre (SMHI) | RCA4 | Tamoffo et al., 2019 |

## 2.2 WRF simulation data

Moreover, we incorporated simulation results from Warscher et al. (2019), who conducted simulations with the Weather Research and Forecast (WRF) model (Skamarock et al., 2008). These WRF simulation results are based on a comprehensive

search and final identification of optimal model physics and parameterization configuration (Wagner and Kunstmann, 2016). They applied a three-domain nested approach with a parent-grid-ratio of 1:3 and a horizontal grid resolution of 5 km for the innermost domain, which frames Germany and the near surroundings. The data used in this study is from their ERA-Interim forced reanalysis run and covers the period 1980 – 2009. Table 2 gives an overview of the physics schemes used in this run as well as in the EURO-CORDEX runs and further information. For more details regarding the model setup, see Wagner and

Kunstmann (2016) and Warscher et al. (2019). As mentioned above, we also used the data from the second domain for some sections, which is of 15 km grid spacing. For this reason, we will refer to WRF@5 km and WRF@15 km from here on to distinguish between the two domains.





## 2.3 Observation data

As reference we used the gridded observational data set from E-OBS (Haylock et al., 2008), version 23.1e, in 0.1° (≈ 11.1
km) horizontal grid resolution. The data contains daily values of the relevant meteorological variables and covers the entire
European land area.

We focused on Germany and its near surroundings as study region from 6° to 15° E and 47° to 55° N. The WRF and E-OBS
data sets were regridded using bilinear interpolation to adjust them to the horizontal grid resolution of the EURO-CORDEX
RCMs.

## 3 Methods

### 3.1 Analysis of precipitation and temperature reproduction

Precipitation and temperature are the main meteorological variables determining droughts. Thus, prior to the drought index
calculation, we analyzed these variables in every RCM and compared them to the reference using Taylor diagrams (Taylor,
2001). They provide a concise visual statistical summary regarding the agreement between patterns in terms of their
correlation, their root-mean-square difference, and the ratio of their variances or standard deviations (Taylor, 2001).

### 3.2 Drought Index: SPEI

There is a variety of drought indices for analyzing different drought characteristics. For the proper selection of a drought
index, its main features like the calculation procedure, input variables, advantages and weaknesses need to be considered
(García-Valdecasas Ojeda et al., 2017). The Standardized Precipitation Index (SPI), developed by McKee et al. (1993), is
one of the most widely used drought indices and recommended by the WMO because of its simplicity, robustness, easy
interpretation, and especially for its multiscalar character. It is comparable across different regions and climates and
therefore very suitable for drought detection around the globe (García-Valdecasas Ojeda et al., 2017). Since precipitation is
the only input variable for the calculation, a high variability is assumed, while other variables like temperature, surface wind
and potential evapotranspiration (PET) are considered as temporal stationary. Thus, the SPI does not define droughts based
on the water balance (Diasso and Abiodun, 2017).





**Table 2.** Overview of the number of vertical levels and model physics schemes.

| Model | Levels | Radiation | Convection | Microphysics | PBL | Land-surface |
|---|---|---|---|---|---|---|
| COSMO-CLM | 40 | Ritter and Geleyn, 1992 | Tiedtke, 1989 | Doms et al., 2011 | Louis, 1979 | Doms et al., 2011 |
| ALADIN 6.3 | 91 | Fouquart and Bonnel, 1980; Mlawer et al., 1997 | Piriou et al., 2007; Guérémy, 2011 | Lopez, 2002 | Siebesma et al., 2007 | Le Moigne, 2012 |
| REMO2015 | 27 | Ritter and Geleyn, 1992 | Tiedtke, 1989 | Lohmann and Roeckner, 1996 | Louis, 1979 | Hagemann, 2002; Rechid et al., 2009 |
| RegCM 4.6 | 23 | Kiehl et al., 1996 | Tiedtke, 1989 | Pal et al., 2000 | Grenier and Bretherton, 2021; Bretherton et al., 2004 | Steiner et al., 2009 |
| RACMO 2.2e | 40 | Fouquart and Bonnel, 1980; Mlawer et al., 1997 | Tiedtke, 1989 | Tiedtke, 1993; ECMWF-IFS, 2007; Tompkins et al., 2007 | Siebesma et al., 2007 | Balsamo et al., 2009 |
| RCA4 | 40 | Savijarvi, 1990 | Bechtold et al., 2001 | Rasch and Kristjansson, 1998 | Cuxart et al., 2000; Lenderink and Holtslag, 2004 | Samuelsson et al., 2015 |
| WRF | 42 | Iacano et al., 2008 | Grell and Freitas, 2014 | Hong and Lim, 2006 | Hong et al., 2006 | Chen and Dudhia, 2001ab |

The Standardized Precipitation Evapotranspiration Index (SPEI), introduced by Vicente-Serrano et al. (2010), overcomes this issue. Because of its dependence on the water balance (precipitation – PET), it incorporates the effects of hot temperatures. That is why it is considered as very useful in terms of global warming (Diasso and Abiodun, 2017; Spinoni et al., 2018). Until today, it has found application in a large number of studies as well (García-Valdecasas Ojeda et al., 2017). We also decided to use the SPEI for this study. In general, the patterns between the SPI and SPEI are usually similar and we

want to take account of the temperature effect, since droughts in the study region predominantly occur in the summer months.





Similar to studies like Diasso and Abiodun (2017), García-Valdecasas Ojeda et al. (2017) and Potopová et al. (2018), the SPEI R Package (Beguería and Vicente-Serrano, 2013) was used for the index calculation. As mentioned above, the SPEI needs PET as additional input variable to precipitation. PET was calculated based on the modified Hargreaves equation
(Droogers and Allen, 2002). The method corrects the PET calculated by the Hargreaves equation by using the monthly rainfall amount as a proxy for insolation and based on the hypothesis, that this amount can change the humidity levels (Vicente-Serrano et al., 2014). By using this method, the PET values are similar to those obtained from the Penman-Monteith method (Allen et al., 2006). The Penman-Monteith method is adopted and recommended by the Food and Agriculture Organization to approximate PET (García-Valdecasas Ojeda et al., 2017), but the variables required for this
method are only included in a limited number of CORDEX simulations. The modified Hargreaves method only requires the maximum and minimum temperatures, so that it is applicable to all data sets used in this study.

For the SPEI calculation, the monthly values of the water balance are used. The obtained time series are fitted to log-logistic distribution. Then the quantiles of the distributions are transformed into standard normal variables. This ensures comparability of the index values across different regions. Negative values indicate drier, positive values wetter than median
conditions (Meresa et al., 2016). To categorize droughts, we follow the most popular classification scheme of McKee et al. (1993) (Table 3).

**Table 3.** Drought classification using SPI/SPEI according to McKee et al. (1993).

| SPI/SPEI value | Drought category |
| --- | --- |
| 0 to -0.99 | Mild |
| -1.00 to -1.49 | Moderate |
| -1.50 to -1.99 | Severe |
| $\leq$ -2.00 | Extreme |

Different aggregation scales for the SPI/SPEI calculation are usually used to define the type of drought. Short time scales up to 3 months are used for meteorological droughts, medium scales of around 6 months for agricultural droughts and longer time scales of 12 months or more refer to hydrological droughts (Wilhite and Glantz, 1985; Heim, 2002; Spinoni et al., 2020). We selected the three months aggregation scale to focus on meteorological droughts. For this reason we will refer from here on to SPEI-3.

SPEI-3 time series were computed for each EURO-CORDEX simulation, the WRF output and the E-OBS reference data set for every grid cell. Correlation analysis between the RCM and reference time series have been conducted as well as a comparison of the index values for the drought event in 2003.

Several metrics are available to assess the spatial agreement between patterns of single RCMs and the reference. Here we used the spatial efficiency (SPAEF) metric (Demirel et al., 2018; Koch et al., 2018). The SPAEF is a multiple components
performance metric developed for the comparison of spatial patterns. While it was originally contemplated for hydrological studies, Koch et al. (2018) state that it is suitable and beneficial for other modelling disciplines too. Correlation, coefficient



of variation as representation for spatial variability and the histogram overlap are used. The three components are independent of each other and equally weighted, so that they complement each other in a useful way and provide holistic pattern information. The SPAEF assesses global characteristics like distribution and variability instead of exact values at the

grid scale. It has a predefined range between -∞ and 1, where 1 corresponds to ideal agreement between two patterns (Koch et al., 2018). For more information regarding the SPAEF and its calculation procedure, see Demirel et al. (2018) and Koch et al. (2018). The metric was used to evaluate spatial agreement for the 2003 event and for the drought characteristics.

**3.3 Drought trend analysis**

To investigate the temporal characteristics of droughts we used the non-parametric Mann-Kendall trend test approach

(Mann, 1945; Kendall, 1975) to detect significant monotonic trends in the index time series at a significance level of 0.05. This approach is based on the correlation between the ranks of a time series and their time order and is commonly used in time series of environmental, climatological or hydrological data (Hamed, 2008; Alhaji et al., 2018). We only considered independent, non-overlapping data.

For a time series $x_1, x_2, x_3 \ldots x_n$, the Mann-Kendall test statistic $S$ is given by

$$S = \sum_{i=1}^{n-1} \sum_{j=i+1}^{n} sign(x_j - x_i) \tag{1}$$

with

$$sign(x_j - x_i) = sign(R_j - R_i) = 1 \text{ if } x_j - x_i > 0 \tag{2}$$

$$sign(x_j - x_i) = sign(R_j - R_i) = 0 \text{ if } x_j - x_i = 0 \tag{3}$$

$$sign(x_j - x_i) = sign(R_j - R_i) = -1 \text{ if } x_j - x_i < 0 \tag{4}$$

where *sign* represents an indicator function, $n$ the number of data points and $R_i$ and $R_j$ their respective ranks. A positive $S$-statistic indicates an increasing trend, a negative one indicates a decreasing trend. When independent data with identical distribution is assumed, the mean and variance of the $S$-statistic is given by

$$\mu = 0 \tag{5}$$

$$\sigma^2 = n(n-1)(2n+5) / 18 \tag{6}$$

Tied ranks or groups (a set of sample data with the same value) reduce the variance to

$$\sigma^2 = \frac{\left\{ n(n-1)(2n+5) - \sum_{j=1}^{p} t_j (t_j - 1)(2t_j + 5) \right\}}{18} \tag{7}$$

where $p$ denotes the number of tied groups and $t_j$ the number of data points in the $j$-th group. Trend significance is tested by comparing the standardized variable $u$ with the standard normal variate at the chosen significance level, where the subtraction or addition of unity is a continuity correction (Kendall 1975; Hamed 2008):

$$u = (S - 1)/\sigma \text{ if } S > 0 \tag{8}$$

$$u = 0 \text{ if } S = 0 \tag{9}$$

$$u = (S + 1)/\sigma \text{ if } S < 0 \tag{10}$$



### 3.4 Analysis of drought characteristics

Drought events and their characteristics have been defined in several ways in the past (Um et al., 2017). Using the SPEI time
series values on the grid point scale, we detected drought events and their characteristics (frequency, duration and severity)
by applying the run theory proposed by Yevjevich (1967), which has been widely employed in drought related studies (e.g.
Spinoni et al., 2014; Marcos-Garcia et al., 2017; Peres et al., 2020; Spinoni et al., 2020). A drought event starts when the
SPEI value falls below -1 for at least two consecutive months. The event ends when the index value returns to positive
values. Drought frequency then describes the number of drought events in a given time period. Drought duration corresponds
to the number of months between start and end of an event (last month not included). Drought severity of an event equals the
sum, in absolute values, of all the monthly SPEI values during the event (Spinoni et al., 2020).

We determined the drought frequency for every grid cell for the whole study period 1980 – 2009. Drought frequencies
between the single grid cells differ and since drought duration and severity refer to every single drought event, we calculated
the mean values for duration and severity for every grid cell to enable a comparison between the single data sets.

### 4. Results and Discussion

### 4.1 Precipitation and Temperature

Figure 1 presents the Taylor diagrams of the grid cell based monthly values of precipitation, maximum ($T_{max}$) and minimum
($T_{min}$) temperature. Here we also added the information of the WRF@15 km dataset to check, if potential WRF benefits are
related to increased resolution or model settings. Regarding precipitation, the WRF@5 km run has the highest correlation
with the reference, it is the only one crossing the 0.75 threshold with a relatively small RMSE score, resulting in the best
overall performance compared to the other RCMs. However, the lowest RMSE is found for RACMO which also holds for
the standard deviation, while WRF@5km, ALADIN, and RCA4 deviate most. Interestingly, the WRF@15 km run has the
lowest correlation and highest RMSE values, while its standard deviation is among the closest to the observational one. This
means that the increased resolution of WRF@5 km leads to improvements in correlation and RMSE scores, but the temporal
variability is better captured in the coarser resolution. The $T_{max}$ Taylor diagram clearly shows a benefit of both WRF runs, so
that we conclude the model setup as the determining factor for the better performance compared to the EURO-CORDEX
RCMs. This is underlined by the fact that the WRF@15 km run has a higher correlation, lower RMSE and matches the
reference standard deviation compared to its 5 km counterpart. Only the two WRF runs reach correlation coefficients above
0.99. Here, all EURO-CORDEX RCMs perform on a similar level, which is high. They all reach correlation values above
0.95 and RMSE below 5. RACMO stands out in this case because of the most accurate standard deviation ratio with the
reference. In the $T_{min}$ Taylor diagram it is obvious that the 5 km WRF run performs best. It has the highest correlation value
(above 0.98), the lowest RSME and it is close to the reference standard deviation. Only the 15 km WRF run is closer in this
regard. The 15 km WRF run and the EURO-CORDEX perform on a similar level. Similar to $T_{max}$, the main difference is the


standard deviation ratio when compared to the reference. In this regard, RACMO has the biggest distance. For $T_{min}$ it seems

like the model setup of WRF leads to benefits compared to the other RCMs, and that the increased resolution brings

additional benefit. From the Taylor diagrams we can conclude that especially $T_{max}$ and $T_{min}$ are very well captured by all

RCMs. There are benefits of increased resolution for precipitation and for $T_{min}$, while for $T_{max}$ mainly the model setup of the

WRF runs is beneficial. The WRF@5 km run performs relatively well in all three variables.

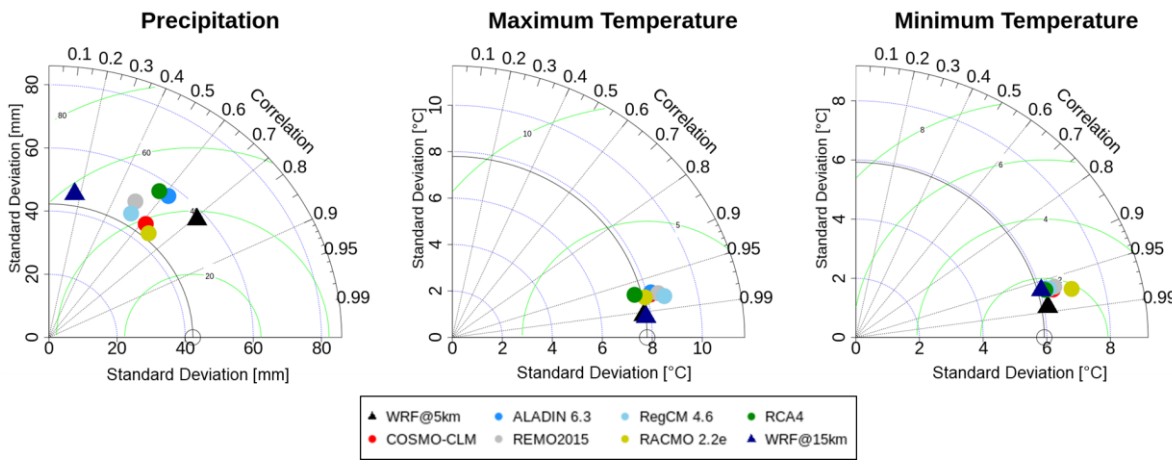


**Figure 1.** Taylor diagrams comparing the model performances in reproducing the monthly values of the meteorological variables in relation to the E-OBS reference data for the study period 1980 – 2009 and the whole study area.

Regarding the resolution effect on the precipitation reproduction, our results are in accordance with findings from, e.g., Tripathi and Dominguez (2013) and Prein et al. (2016), who found that higher resolution leads to better reproduction. Our

results are in contrast to findings from, e.g., Rauscher et al. (2010), Casanueva et al. (2016) and Dieng et al. (2017), who could not identify a benefit of increased resolution for both, general pattern and on annually mean basis. It must be noted though, that in all the studies mentioned the differences between the two resolutions analyzed were much bigger than in our case, whereas both resolutions (12.5 km and 5 km) are usually already considered as high-resolution in the literature. In the studies mentioned, there is always a resolution of 50 km compared to 25 km (Rauscher et al. (2010)), 12.5 km (Casanueva et

al., 2016; Prein et al., 2016; and Dieng et al., 2017), and 10 km (Tripathi and Dominguez, 2013). Therefore, it can be assumed that if existent, the benefits of a resolution increase from 5 to 12.5 km are less distinct. One must also keep in mind that the studies were conducted in different regions, which certainly plays a role too and that often different resolutions of the same RCM were compared. The results from this section further show that RCMs with reasonable performance in simulating one or both temperature variables do not necessarily reproduce precipitation as good, which is in accordance to

findings from Peres et al. (2020). They further found that COSMO-CLM and RACMO showed good performance in reproducing precipitation, while RCA4 and WRF struggled the most. Regarding mean temperature, COSMO-CLM and REMO showed best performances, RCA4, ALADIN and RACMO the worst. This could in part be confirmed by the results here for the precipitation reproduction: COSMO-CLM and RACMO perform relatively well while RCA4 showed a


relatively poor performance. It must be noted that Peres et al. (2020) analyzed the mean temperature instead of $T_{max}$ and $T_{min}$
and that they looked at different temporal scales. Moreover, they employed EURO-CORDEX RCMs with different GCMs as
forcing, while here all RCMs had the same ERA-Interim forcing.

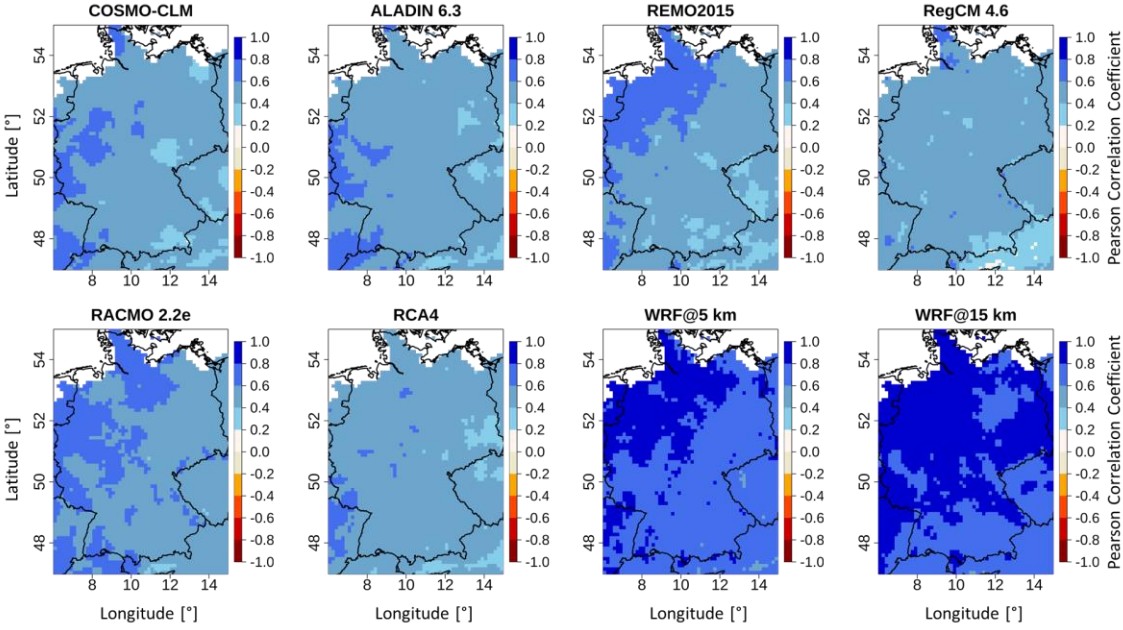

**Figure 2.** Grid cell based Pearson Correlation Coefficients of the SPEI-3 time series for 1980 – 2009 between each RCM
and E-OBS.

**4.2 SPEI Time Series Correlation**

Figure 2 shows the grid cell based Pearson correlation coefficients of the SPEI-3 time series between the E-OBS reference
data and the single RCMs. Here we show both the WRF@5 km and the WRF@15 km run. The domain mean values of the
Pearson correlation coefficient are given in Table 4. It is evident that the two WRF runs have higher correlation values all
over the domain compared to the EURO-CORDEX RCMs. This could be expected from the results in the Taylor diagrams
(Figure 1) and is further corroborated by the values in Table 4. Only the WRF runs cross the 0.7 threshold, while none of the
other RCMs even exceed the 0.6 threshold. RegCM holds the lowest mean correlation value (0.48). Generally, there is a
similarity in the spatial patterns of the EURO-CORDEX RCMs: most of the domains are covered by values between 0.4 and
0.6. Especially in REMO and RACMO there are areas in the western part of the domain with higher values (0.6 – 0.8).
Interestingly, this is also the case in the WRF@5 km domain, here with values ranging between 0.8 and 1. It is clearly visible
that the WRF@15 km run outperforms the WRF@5 km run and thereby has the overall best performance. This indicates that
the WRF benefits can be attributed to the WRF model settings and not to the increased resolution.





**Table 4.** Domain mean values of the Pearson Correlation Coefficient from each RCM in Figure 2.

| Model | Mean Correlation Coefficient |
|---|---|
| COSMO-CLM | 0.51 |
| ALADIN 6.3 | 0.50 |
| REMO2015 | 0.52 |
| RegCM 4.6 | 0.48 |
| RACMO 2.2e | 0.57 |
| RCA4 | 0.50 |
| WRF@5 km | 0.77 |
| WRF@15 km | 0.80 |


### 4.3 Drought Event August 2003

In the following the SPEI-3 scores for the drought event in August 2003, one of the major drought events in central Europe in the last decades (e.g. Fink et al., 2004; Rebetez et al., 2006; Ionita et al., 2021), are analyzed. Because of the results in the previous section, here we focus on the values from the WRF@5 km run in direct comparison to the reference values from E-
OBS (Figure 3). Relevant scores of the other RCM runs are given in Table 5.

The E-OBS spatial pattern reveals that especially the southern half of the domain was mostly under extreme (SPEI ≤ -2, see Table 3) drought conditions, while in the northern half moderate to severe (-1 to -2) drought conditions were predominant. This pattern is not well reproduced by WRF@5 km, which is also reflected by the low SPAEF value (0.21) in Table 5. The WRF@5 km domain is predominated by values between -1 and -2, so the biggest accordance with E-OBS can be found in
the northern half. Some areas of the domain range between 0 and -1, indicating mild drought conditions. Punctually, there are some spots with extreme drought values as well, which do not match with E-OBS values though. In both domains the entire area is covered nearly only with negative values, which underlines the distinct drought conditions of that period. The mean SPEI-3 values in Table 5, which are all negative, further confirm this. It is striking that E-OBS holds the lowest mean value (-1.90), which corresponds to severe drought conditions. The highest mean value is hold by RACMO (-0.97),
corresponding to mild drought conditions. This highlights the big differences among the RCMs. The percentage of area under drought (AUD) is defined as the percentage of grid cell with values of ≤ -1 in relation to the total number of grid cells. Here we see distinct differences between the single RCMs and the reference. While E-OBS and RCA4 have AUD values of more than 80 %, these values are even below 50 % in REMO and RACMO. These two RCMs also hold the two highest mean bias values (-1.03 and -0.93 SPEI units). RCA4 holds the lowest mean bias value (-0.18 SPEI units). All mean bias
values are negative, which is a further indication of the drought underestimation of the RCMs. The SPAEF values are either negative or very low. The only exception is ALADIN with the maximum value of 0.55. REMO hold the lowest SPAEF value (-1.89), which completes the overall bad performance this RCM in this regard.

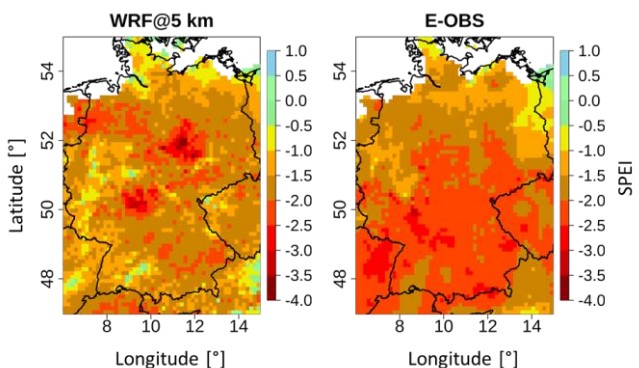

**Figure 3.** SPEI-3 values for August 2003 from WRF@5 km and E-OBS.

It can be concluded that there are distinct differences between the single RCM performances regarding the reproduction of single drought events. None of the RCMs was able to satisfactorily reproduce the spatial patterns of the reference. Also, the correct representation of the mean drought index values and the AUD values turned out to be difficult. Thus, the results confirm findings from Um et al. (2017), who found that the spatial extents of droughts diverge among the RCMs and that the RCMs are not able to accurately capture drought events with large spatial scales. Since WRF@5 km did not perform best in

any of the categories, there does not seem to be any benefit of increased model resolution and model settings in this regard. Moreover, it is striking that the three RCMs with the highest mean bias values and the biggest differences from E-OBS in domain mean value and AUD, COSMO-CLM, REMO and RACMO, were running with the Tiedtke (1989) convection scheme. Especially the scheme configurations of COSMO-CLM and REMO are similar (Table 2).

**Table 5.** Drought Event August 2003 - SPEI-3 metrics including the Spatial Efficiency (SPAEF) scores for the spatial agreement between each RCM and E-OBS as reference.

| Model | Mean SPEI-3 | Area under Drought [%] | Mean Bias [SPEI units] | SPAEF |
|---|---|---|---|---|
| COSMO-CLM | -1.18 | 66.0 | -0.72 | -0.01 |
| ALADIN 6.3 | -1.29 | 69.3 | -0.60 | 0.55 |
| REMO2015 | -0.89 | 43.6 | -1.03 | -1.89 |
| RegCM 4.6 | -1.35 | 67.5 | -0.56 | -0.53 |
| RACMO 2.2e | -0.97 | 45.1 | -0.93 | 0.03 |
| RCA4 | -1.73 | 81.8 | -0.18 | 0.07 |
| WRF@5 km | -1.61 | 78.0 | -0.27 | 0.21 |
| E-OBS | -1.90 | 81.7 | | |




## 4.4 SPEI Trend Analysis

Figure 4 displays the results of the Mann-Kendall trend test for all RCMs and the reference for the SPEI-3 time series of
each grid cell. It is important to note that the Mann-Kendall trend test gives information about whether there is a monotonic
positive, negative or no trend in a time series at a certain level of significance (here 0.05). There is no information about
exact trend values.

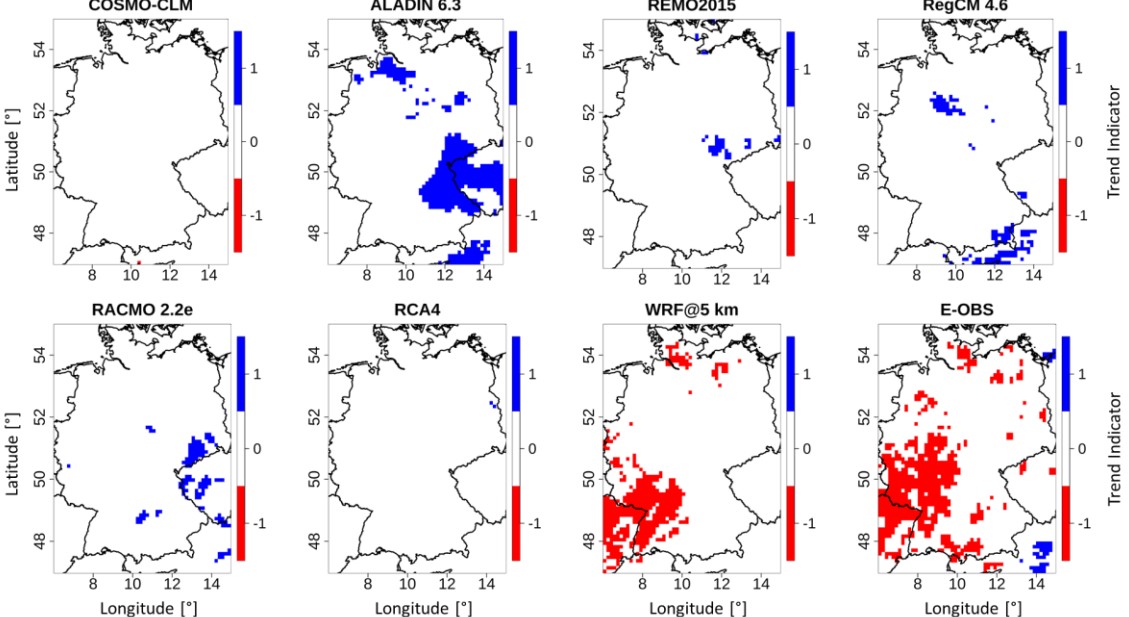

**Figure 4.** Grid cell based SPEI-3 trends for 1980 – 2009 based on the Mann-Kendall Test for each RCM and E-OBS.

It is striking that only WRF@ 5km is able to reproduce negative trend signals which are also existent in the references and
indicate a drying trend. None of the EURO-CORDEX RCMs is able to reproduce this. In the WRF@5 km domain, the
locations of the negative trends are even locally represented accurately, concentrated mainly in the southwestern parts of the
domain and partly in northern regions. These findings could be inferred from the results in Sect. 4.2. Most of the domain
area of each RCM and the reference shows no trend (Table 6). If there is a trend in the EURO-CORDEX RCMs, it is always
positive, indicating increasing SPEI-3 values and thus wetter conditions. This is the case for ALADIN, REMO, RegCM and
RACMO. COSMO-CLM and RCA4 show almost entirely white domain areas. Interestingly, the E-OBS domain has only
small parts of positive trend areas, concentrated in the southeastern corner and partly in northeastern parts. There is only
slight agreement in ALADIN, RegCM and RACMO in this regard. The WRF@5 km domain shows no positive trend grid
cells at all (Table 6).
To answer the question whether the agreement of WRF and E-OBS regarding the negative trend areas is due to the increased
resolution or to the model settings, we applied the Mann-Kendall trend test also to the WRF@15 km run (Figure 5). There is
clear indication that the reproduction is not primarily linked with the increased resolution since the negative trends are



represented here too. Compared to the WRF@5 km runs, the negative trend areas are much more spacious. This is also reflected in Table 6: more than one third of the domain (34.2 %) is covered by negative index values, which is more than

double compared to E-OBS (16.9 %) and more than three times compared to WRF@5 km (10.8 %), underlining the big overestimation of negative trend areas. There are no positive trend values in the WRF@15 km domain either.

**Table 6.** SPEI-3 Trends overall metrics.

| Model | negative [%] | neutral [%] | positive [%] |
|---|---|---|---|
| COSMO-CLM | 0.03 | 99.97 | 0 |
| ALADIN 6.3 | 0 | 87.3 | 12.7 |
| REMO2015 | 0 | 98.98 | 1.02 |
| RegCM 4.6 | 0 | 96.7 | 3.3 |
| RACMO 2.2e | 0 | 96.97 | 3.03 |
| RCA4 | 0 | 99.95 | 0.05 |
| WRF@5 km | 10.8 | 89.2 | 0 |
| E-OBS | 16.9 | 81.9 | 1.2 |
| WRF@15 km | 34.2 | 65.8 | 0 |

From this section it is concluded that there are clear benefits of the WRF runs in the appropriate trend reproduction. As seen, these benefits are not primarily due to increased resolution, but to the model settings, highlighting the high importance of model configurations tailored to the target region. However, the increased resolution brings further benefits and leads to higher agreement with the reference. The EURO-CORDEX RCMs completely fail in this aspect.

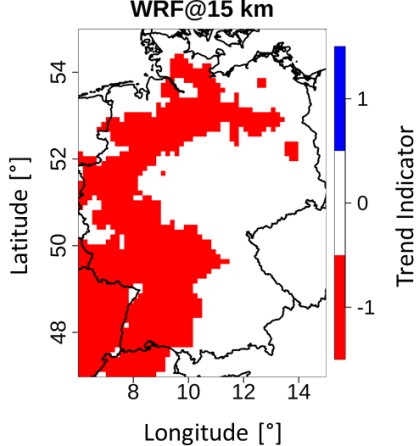

**Figure 5.** Grid cell based SPEI-3 trends for 1980 – 2009 based on the Mann-Kendall Test for WRF@15 km.




## 4.5 Drought Characteristics Analysis

### 4.5.1 Drought Frequency

Figure 6 presents the E-OBS drought frequency pattern for the time period 1980-2009 based on the SPEI-3 along with the grid cell based differences between each RCM and E-OBS. Table 7 gives more detailed information. As described in Sect.

3.4, we considered a drought event as such when the SPEI value falls below -1 for at least two consecutive months. So already moderate droughts (see Table 3) are taken into account. The drought frequency gives the number of drought events in the given time period for each grid cell.

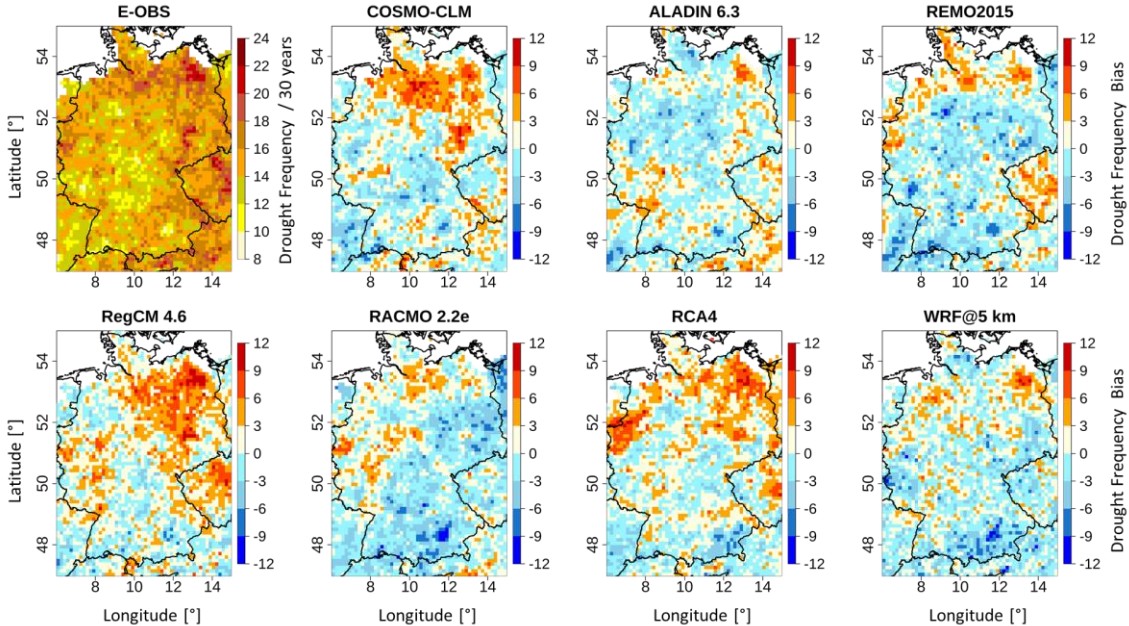

**Figure 6.** Grid cell based E-OBS drought frequency pattern based on the SPEI-3 between 1980 – 2009
and differences between each RCM and E-OBS.

The meteorological drought frequency pattern in E-OBS shows that every single grid cell experienced at least eight drought events within the 30 years timespan. The mean value for the whole domain is 15.5 (Table 7). The highest number of droughts occurred in the northeastern part with some grid cells reaching values of up to 24. Generally, the eastern half of the domain has higher values and towards the southwest the number of drought events decreases. The RCM difference patterns

differ among each other. Relatively high positive bias values (between 3 and 12) are often found in the northern and northeastern parts of the domain, especially in COSMO-CLM, RegCM and RCA4. The southern half of the domain is rather predominated by negative bias values in all RCMs. There is a similarity between the patterns of ALADIN and WRF@5 km. All in all, bias values of ±9 are rare in all RCMs, the major part of the RCM domains rather ranges between ±6. For the drought characteristics we only considered absolute values for the mean bias calculation (third column in Table 7) since

values with opposite signs can balance each other out, thus making the information less meaningful. RCA4 holds by far the





smallest bias value (0.70), while the values of the other RCMs are very close to each other with RegCM holding the maximum (2.95). A look at the domain mean number of drought events per 30 years shows that there is no big difference between the single values. They mainly range around 15, indicating that in average every second year in the considered time period a meteorological drought event has taken place. RegCM (13.4) has the biggest difference (2.1) to the reference,

WRF@5 km and REMO the smallest (0.1). This and the mean bias values speak for reasonable performances of the RCMs regarding the reproduction of the mean frequency conditions. The SPAEF values give information about the pattern agreement between the reference and the individual RCMs (not shown here). It is striking that all values are negative, which indicates there is no good overall spatial agreement at all. COSMO-CLM holds the lowest value (-0.38), RACMO the highest (-0.09).

**Table 7.** Drought Frequency SPEI-3 metrics.

| Model | Mean<br>[n Events / 30 years] | Mean Bias<br>[n Events] | SPAEF |
|---|---|---|---|
| COSMO-CLM | 14.3 | 2.61 | -0.38 |
| ALADIN 6.3 | 14.9 | 2.05 | -0.27 |
| REMO2015 | 15.4 | 2.44 | -0.14 |
| RegCM 4.6 | 13.4 | 2.95 | -0.28 |
| RACMO 2.2e | 15.9 | 2.37 | -0.09 |
| RCA4 | 13.9 | 0.70 | -0.34 |
| WRF@5 km | 15.6 | 2.23 | -0.14 |
| E-OBS | 15.5 | | |

From this section it is concluded that there is no benefit of WRF's increased resolution and model setup regarding the reproduction of the drought frequency. In fact, all the RCMs performed on a similar level. Furthermore, the mean conditions of the drought frequencies are sufficiently well reproduced. The focus should therefore be put on the information retrievable

from the mean conditions and not on spatial accuracy.

**4.5.2 Mean Drought Duration**

Figure 7 shows the SPEI-3 based mean drought duration pattern for the period 1980-2009 from E-OBS and the grid cell based differences with the RCMs.



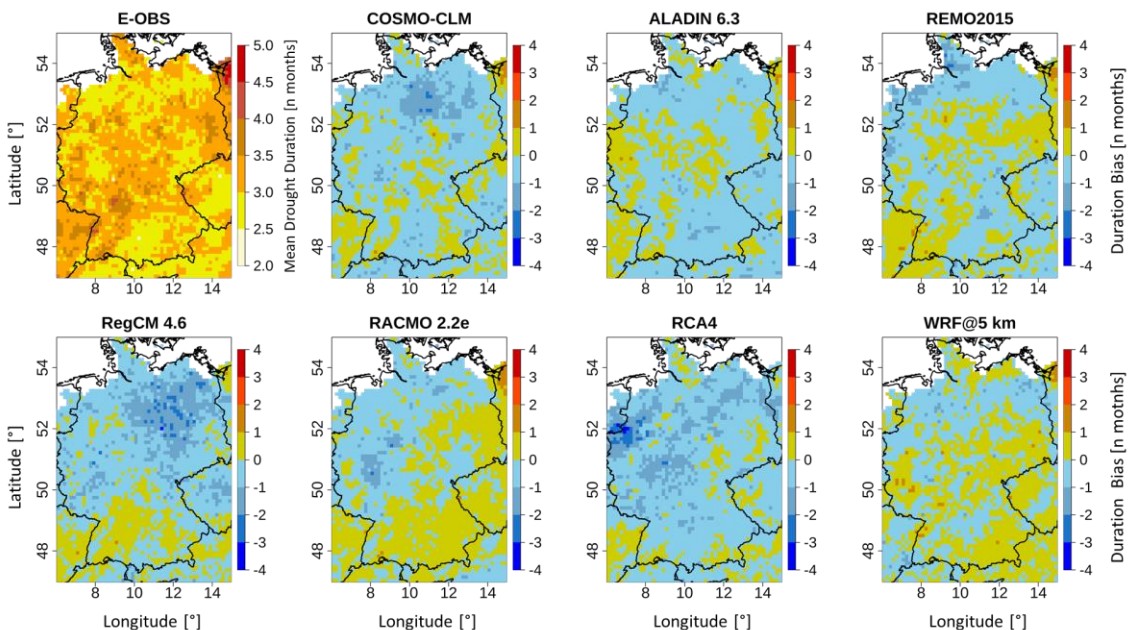

**Figure 7.** Grid cell based E-OBS mean drought duration pattern based on the SPEI-3 between 1980 – 2009 and differences
between each RCM and E-OBS.

**Table 8.** Mean Drought Duration SPEI-3 metrics.

| Model | Mean [n months] | Mean Bias [n months] | SPAEF |
|---|---|---|---|
| COSMO-CLM | 3.5 | 0.45 | 0.01 |
| ALADIN 6.3 | 3.4 | 0.36 | 0.07 |
| REMO2015 | 3.3 | 0.40 | -0.01 |
| RegCM 4.6 | 3.6 | 0.57 | -0.11 |
| RACMO 2.2e | 3.2 | 0.35 | 0.09 |
| RCA4 | 3.7 | 0.58 | 0.09 |
| WRF@5 km | 3.1 | 0.34 | 0.20 |
| E-OBS | 3.1 | | |

The E-OBS mean meteorological drought duration pattern is quite uniform, almost the entire domain is covered by values
ranging between 2.5 and 3.5 months. The domain mean value (3.1 months) in Table 8 underlines this. The vast majority of
the RCM bias domains is covered by values between 2 and -2 months, implying some similarities between single RCMs.
One thing all RCMs have in common is that the northern parts are predominated by negative bias values. COSMO-CLM,
RegCM and RCA4 are predominated by negative bias values almost all over their entire domains. Table 8 shows that all
mean bias values are below 1 month, with WRF@5 km holding the lowest value (0.34 months) and RCA4 the highest (0.58
months). The RCM domain mean drought durations are all equal or higher than the reference value (3.1 months) with





WRF@5 km being closest and RCA4 being furthest away (0.6 months). As inferred by the maps and mean values, the SPAEF values between the reference and the single RCM patterns (not shown) are higher compared to the drought frequency values (section above). Only REMO and RegCM hold negative values. WRF@5 km has by far the highest value (0.20), which is still relatively low despite everything, while the other RCMs do not cross the 0.1 threshold.

It is concluded that WRF has no real benefit due to increased resolution or model setup. The benefit is perhaps somewhat present regarding the spatial agreement with the reference, but although the SPAEF achieved by WRF@5 km is distinctly higher than that from the EURO-CORDEX RCMs, it is still not reliable. Nevertheless, as for the drought frequencies in the section above, all RCMs provide a satisfying reproduction of the mean conditions. Here, there is also a lack of spatial accuracy, but this deficiency is less pronounced.

### 455 4.5.3 Mean Drought Severity

Figure 8 displays the E-OBS SPEI-3 based mean drought severity pattern for the time period 1980-2009 and the grid cell based differences with the RCMs.

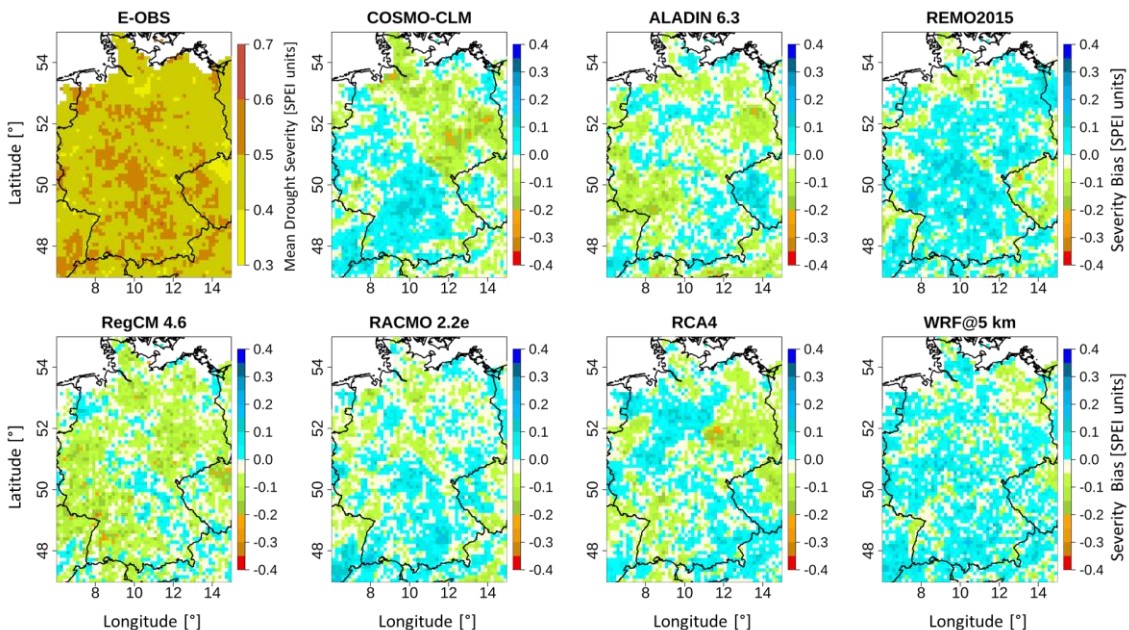

**Figure 8**. Grid cell based E-OBS mean drought severity pattern based on the SPEI-3 between 1980 – 2009 and differences
between each RCM and E-OBS.

The E-OBS domain shows a pretty uniform pattern with the majority of the values ranging between 0.4 and 0.5 SPEI units. The domain mean value (0.47 SPEI units) in Table 9 confirms this. This value further implies that, if all droughts beginning from a SPEI value of -1 are considered, the mean severity is -1.47 SPEI units. This means that the mean drought severity can 465 still be classified as moderate according to Table 3, but it is very close to severe threshold. In general, all RCMs show





overall low bias values, which is also displayed in the mean bias values in Table 9: the maximum mean bias value is 0.07 SPEI units and is held by RegCM. Especially RACMO and WRF@5 km show domains with only a few dark color shaded areas, which is also reflected in the lowest mean bias values (0.04 SPEI units). Considering all RCM domains, it is not possible to determine areas of preferably positive or negative bias values, as the same areas have different signs in different 
RCMs. Neither it is possible to determine regions of preferably high bias values across all RCMs. The domain mean severity values are very close to each other, all around 0.5±0.04 SPEI units with a range of 0.07 SPEI units between maximum (RegCM) and minimum (REMO and WRF@5 km). Regarding the spatial agreement between E-OBS and the single RCMs (not displayed here), there are again overall low SPAEF values, pointing towards a low level of agreement. WRF@5 km holds the highest value (0.14) and is the only one exceeding the 0.1 threshold. COSMO-CLM holds by far the lowest value (-
0.15). The values of ALADIN and RCA4 are also negative.

**Table 9.** Mean Drought Severity SPEI-3 metrics.

| Model | Mean [SPEI units] | Mean Bias [SPEI units] | SPAEF |
|---|---|---|---|
| COSMO-CLM | 0.50 | 0.06 | -0.15 |
| ALADIN 6.3 | 0.50 | 0.05 | -0.02 |
| REMO2015 | 0.46 | 0.05 | 0.03 |
| RegCM 4.6 | 0.53 | 0.07 | 0.02 |
| RACMO 2.2e | 0.48 | 0.04 | 0.09 |
| RCA4 | 0.49 | 0.05 | -0.04 |
| WRF@5 km | 0.46 | 0.04 | 0.14 |
| E-OBS | 0.47 | | |

Similar to the two previous sections, it is concluded that the mean drought severity conditions are captured reasonably well 
by the RCMs in terms of domain mean values, while the spatial accuracy is overall not satisfying. Regarding the former, all RCMs perform on a similar level. This means that there is no benefit of WRF due to its increased resolution or model setup detectable in this regard here either. Peres et al. (2020) found that the RCMs with the best performance for precipitation mostly performed well regarding the reproduction of drought characteristics, too. This cannot really be confirmed here in our findings. As stated in Sect. 4.1, COSMO-CLM and RACMO perform overall especially well for precipitation. Regarding the 
drought characteristics, these two RCMs could not stand out overall. Only in some aspects there were marginal benefits. It must be noted that Peres et al. (2020) used another methodology regarding the definition and calculation of drought characteristics, since they worked with precipitation threshold values instead of drought indices.

From an overall perspective, it can be stated that no specific physics scheme of the RCMs (Table 2) considered on its own turned out to be superior to the others for the reproduction of the drought characteristics.





## 5. Conclusions

A drought analysis for Germany and the near surroundings for the period 1980-2009 is conducted in this study. We address the influence of increased model resolution and appropriate model configuration on the reproduction of the SPEI drought index for the three months aggregation scale. For that purpose, an ensemble of six ERA-Interim driven EURO-CORDEX RCMs of 12.5 km horizontal grid resolution and an ERA-Interim driven high-resolution (5 km) WRF run, whose setup was tailored to the target area, are employed. The outputs are evaluated regarding their ability to reproduce precipitation, $T_{max}$ and $T_{min}$ as well as SPEI-3 based correlations and trends, the drought event in 2003 and overall drought characteristics (frequency, duration and severity). E-OBS data serves as reference.

WRF with its increased resolution and tailored model setup is shown to be not beneficial regarding the reproduction of the drought event 2003 and the overall drought characteristics. Despite the same forcing, the RCMs exhibit a large spread. The drought event 2003 is not well captured by any RCM. As for the domain mean conditions of the overall characteristics, they are reasonably well reproduced in all cases. The spatial agreement with the reference, though, is not satisfactory for any RCM. This is especially the case for the drought frequencies. Meteorological droughts are found to occur approx. 16 times in the study period with an average duration of 3.1 months and average severity of 1.47 SPEI units. No specific physics scheme or configuration can be shown to be especially beneficial for the reproduction of the drought characteristics. These results suggest that, depending on the goal in drought analysis, a resolution of 12.5 km may be sufficient to get to similar findings as with higher resolutions. This can save computation resources. WRF's increased resolution and setup is turned out to be beneficial in the analysis of the monthly values of the meteorological variables and the correlations of the SPEI time series. The latter can primarily be attributed to the model setup. However, the greatest benefit of WRF is found in the reproduction of the SPEI trends. It is the only RCM that captures the negative trends of the reference, while all EURO-CORDEX RCMs fail in this aspect. This is primarily due to the better model optimization for the area of interest compared to the larger-extent EURO-CORDEX runs, which highlights the importance of such tailored physics settings. Higher resolution additionally leads to greater spatial accuracy. These findings can be of high relevance, since appropriate reproduction of drought index trends is an important feature of RCMs, especially in the context of climate change analysis. Furthermore, the results may guide in selecting suitable RCMs for certain aspects of drought analysis in Germany and similar regions in a historical context and also for future projections.

## Data Availability

The EURO-CORDEX data is freely available at the EURO-CORDEX website (https://www.euro-cordex.net/). The E-OBS data is freely available at the ECA&D website (https://www.ecad.eu/). The WRF data and the associated configuration files can be obtained online from Petrovic (2022, https://doi.org/10.5281/zenodo.6577187).



## Author contribution

DP, BF and HK developed the methodology for the study. DP carried out the data analysis and drafted the manuscript, with support of BF and HK. HK provided grant funding and supervised the research.

## Competing interests

The authors declare that they have no conflict of interest.

## Acknowledgements

The authors gratefully acknowledge the work of the WRF modeling community, the European Centre for Medium-Range Weather Forecast for the reanalysis data ERA-Interim, the contributors to the EURO-CORDEX projects used in this study, the ECA&D group for the E-OBS data set and Warscher et al. (2019) for providing the WRF simulation data. Great thanks also to Gerhard Smiatek for his support. This work is funded by the ClimXtreme project of the BMBF (German Federal Ministry of Education and Research) under grant "Förderkennzeichen 01LP1903J".

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
