# Peer review of "Droughts in Germany: Performance of Regional Climate Models in Reproducing Observed Characteristics"

_Natural Hazards and Earth System Sciences, 2022_

## Author Comment (AC1)

**Response to RC1 on nhess-2022-162**

NOTE: Reviewer's comments are in black, our responses to the comments are given in blue below.

The main achievement of the analysis performed by Petrovic et al. consists of providing insights into the utility of improving spatial resolution and customizing the model setup of RCMs aimed at reproducing (and, possibly, projecting) drought characteristics. Through a simple yet straightforward experiment, they give clear answers that can drive further development of RCMs aimed at reproducing and projecting drought characteristics. I only have two main suggestions for the authors:

1) I wonder if SPEI behaviour for other aggregation scales would be the same. Would it be possible to give any information (even as supplementary material) also for SPEI-6 and SPEI-12?

> Yes, we agree with the reviewer that this information might be useful to the readers. Thus, we will add supplemental material about SPEI-6 and SPEI-12 and add some text to the respective sections. It will mainly be about the changes depending on the aggregation scale.

2) In the Introduction, the authors declare the objective of gaining insights into drought development for Germany etc., but I can't find specific indications about that in the paper. A detailed section could help; otherwise, I suggest avoiding emphasizing this objective in the introduction.

> Drought development was meant to refer to the meteorological drought development in Germany (and the near surroundings) in the considered time period 1980 – 2009 based on the E-OBS reference data. These results are mainly included in section *4.5 Drought Characteristics Analysis* and the numbers from the respective tables are given in the abstract and in the conclusions. We agree that another term than *development* would be more accurate, so we have decided to use the term *course*: "2. To gain insights into the meteorological drought course for Germany and the near surroundings between 1980-2009."

L84: ERA-Interim, capital I.

> Correct. We will change it.

L87: "concluded". In the line below: "saw".

> Correct. We will change both.

Table 1: are those listed the only models available? Are there others (e.g. HIRHAM)? Please specify in the text. If some models are neglected, please explain why.

> The models listed depict the complete subset for the two requirements: Containing the necessary variables (precipitation, Tmax and Tmin) and coverage of the time period 1980 – 2009. To make this clearer in the text, we will change it to: "At the time of selection, these were all available model runs that cover the study period 1980 – 2009 and contain the relevant meteorological variables needed for the analysis."

L222: to make the paper more self-consistent, I suggest providing more information about how the SPAEF metric works. On the other hand, less room can be devoted to the Mann-Kendall test, which is older and more widely known.

> We can see the point and will provide more information about the SPAEF metric. We will also work on shortening the section with the Mann-Kendall trend test.

Section 4.1: Taylor diagrams don't provide information about possible bias. I suggest adding this piece of information.

> Indeed, there are no bias information included. Therefore, we will add a table with bias values from the spatially and temporally averaged monthly values of the three important SPEI variables. Text for description will be added in the results section and also in the conclusions.

L296: from 50, I guess.

> Thanks for the hint. It should be "from 12.5 to 5 km". We will change the sentence (also after a comment of reviewer 2) to "From our results, we obtain that, if existent, the benefits of a resolution increase from 12.5 to 5 km are less distinct.".

L314: I suggest removing Table 4 and introducing the mean correlation coefficient as an inset in the respective map (or near the title).

> We agree. Table 4 will be removed and the mean correlation coefficient values will be added underneath the title of the respective models.

Section 4.3: why not use WRF@15 km here, at least in Table 5? It would further highlight the benefits of model settings.

> We agree with the reviewer. We will add the map of the WRF@15 km run to Figure 3 and the relevant scores to Table 5. Figure 3 will then contain both WRF runs next to the E-OBS reference. Additional text will be added in this section and in the conclusions section.

Table 5: are the Mean SPEI values and the other statistics averaged over the German territory or the whole domain?

> They were averaged over the whole domain.

L347: … of this RCM …

> Correct. *Of* will be included: "The only exception is ALADIN with the maximum value of 0.55. REMO holds the lowest SPAEF value (-1.89), which completes the overall bad performance of this RCM in this regard."

L355: in the authors' opinion, why do we see these results with the Tiedtke scheme? However, from Table 2, I observe that also RegCM uses Tiedtke.

> We have further analyzed the relevant time steps (months) of the spatially averaged time series from the three meteorological variables to check if the deviations from the reference are also the highest in this case. It turned out that the RCMs, which were run with the Tiedtke convection scheme, do not show extraordinary bias values here. From this we conclude that the deterioration in SPEI-3 performance for the 2003 event is rather not related to the Tiedtke scheme. The fact that RegCM was also run with Tiedtke corroborates this conclusion. Consequently, we will remove the section in question.

Figs. 4 and 5: I would merge them. In general, I suggest always considering as two different configurations WRF@5km and WRF@15km (see my comment to Section 4.3).

> We agree that merging the two figures would make sense, but see the difficulty in the resulting figure dimensions since this would result in nine panels. As a compromise we would like to keep the Figs. 4 and 5 the way they are and add the WRF@15 km scores in the Tables 7 − 9 in the drought characteristics section. Then the WRF@15 km information will be included in every single section. This implies also some adjustments in the results and conclusions section.

Section 4.5.1: E-OBS drought frequency looks too high for some areas (up to 22/24 times over 30 years). What is the reference period on which the index is calculated? 1980-2009?

> We can see that the drought frequency values appear relatively high for some areas. It must be kept in mind that already events with an SPEI-3 value equal to or below -1 are considered as droughts here. This does not necessarily imply drought events to be severe or extreme. Due to the definition of the SPEI, this can also imply a just drier than normal period (dry anomaly), which is then considered as a drought event. This can also happen in autumn and winter months. For clarification, some description for the proper classification will be added to the text.
>
> Yes, the reference period for the index calculation is 1980 – 2009.

L414: so, it is mean absolute error (MAE).

> Correct, it is the mean absolute error. This information will be added to the text: "For the drought characteristics we used the mean absolute error (MAE) as a measure for the domain mean bias (third column in Table 6) since values with opposite signs can balance each other out, thus making the information less meaningful."

Sections 4.4 and 4.5: discussion, specifically in terms of comparison to existing literature, is mostly missing.

> We can see the point here. Searching for literature dealing with similar topics (trend detection and drought characteristics) for the same or similar region was not very fruitful. However, some discussion can be added to section 4.4 regarding model capabilities to reproduce observed trends using the Mann-Kendall trend test: "Nasrollahi et al. (2015) applied the Mann-Kendall trend test to the outputs of 41 CMIP5 models to evaluate their ability to replicate observed drought trends on the global scale between 1901 – 2005. They used the SPI-3 as drought index (and SPI-6 in the supporting material). Their results revealed that about 75 % of the models reproduce the global drying trend, but most models fail at reproducing regional wetting and drying trends (at most about 40 % with agreement). In most locations, less than 10 % of the models showed agreement with the observations. Greater agreement was found in higher latitudes. Um et al. (2017) also performed the Mann-Kendall trend test on grid cell based SPEI-12 time series from outputs of four (HadGEM3-RA, MM5, RegCM4 and RSM) RCMs from CORDEX East Asia and of their ensemble mean for the time period 1980 – 2005 over East Asia. They found distinct differences among the single model outputs regarding their capability to reproduce observed drying and wetting trends. While HadGEM3-RA and MM5 generally captured the proper trends, RegCM4 and RSM were only partially successful. This is why the ensemble mean showed relatively poor performance compared to the two former RCMs. These results highlight the spread in the model's capability in reproducing observed trends of wetting and drying, which is found in this study as well.".
>
> Regarding the drought characteristics, matching literature is sparse. Moreover, different methods, time and aggregation scales, definitions, reference data sets etc. make comparisons with existing literature not really meaningful in most cases.

Conclusions: maybe, the main achievements of the paper can be highlighted with bullet points (e.g., increased resolution and setup are not useful for drought characteristics; they are helpful for correlations; they are useful for trends, etc.).

We thank for the suggestion and comprehend that this would make it very clear. Nevertheless, we would like to keep the format in continuous text. Also, because it appears more common in NHESS.

---

## Author Response (AR1)

**Author's response to the anonymous referee comments on nhess-2022-162**

Dear anonymous reviewers,

we sincerely thank you for the time and effort you took to review our manuscript and for providing constructive feedback and comments that will help improve the quality. The manuscript has been revised according to your suggestions. Please find below our response to each of your comments.

NOTE: Reviewer's comments are in black, our responses to the comments are given in blue below.

**Response to RC1 on nhess-2022-162**

The main achievement of the analysis performed by Petrovic et al. consists of providing insights into the utility of improving spatial resolution and customizing the model setup of RCMs aimed at reproducing (and, possibly, projecting) drought characteristics. Through a simple yet straightforward experiment, they give clear answers that can drive further development of RCMs aimed at reproducing and projecting drought characteristics. I only have two main suggestions for the authors:

1) I wonder if SPEI behaviour for other aggregation scales would be the same. Would it be possible to give any information (even as supplementary material) also for SPEI-6 and SPEI-12?

    Yes, we agree with the reviewer that this information might be useful to the readers. Thus, we have added supplemental material about SPEI-6 and SPEI-12.

2) In the Introduction, the authors declare the objective of gaining insights into drought development for Germany etc., but I can't find specific indications about that in the paper. A detailed section could help; otherwise, I suggest avoiding emphasizing this objective in the introduction.

    Drought development was meant to refer to the meteorological drought development in Germany (and the near surroundings) in the considered time period 1980–2009 based on the E-OBS reference data. These results are mainly included in section *4.5 Drought Characteristics Analysis* and the numbers from the respective tables are given in the abstract and in the conclusions. We agree that another term than *development* would be more accurate, so we have decided to use the term *course*: "2. To gain insights into the meteorological drought course for Germany and the near surroundings between 1980–2009."

L84: ERA-Interim, capital I.

    Correct. We have changed it.

L87: "concluded". In the line below: "saw".

    Correct. We have changed both.

Table 1: are those listed the only models available? Are there others (e.g. HIRHAM)? Please specify in the text. If some models are neglected, please explain why.

    The models listed depict the complete subset for the two requirements: Containing the necessary variables (precipitation, $T_{max}$ and $T_{min}$) and coverage of the time period 1980–2009. To make this clearer in the text, we have changed it to: "At the time of selection, these were all available model runs that cover the study period 1980–2009 and contain the relevant meteorological variables needed for the analysis."

L222: to make the paper more self-consistent, I suggest providing more information about how the SPAEF metric works. On the other hand, less room can be devoted to the Mann-Kendall test, which is older and more widely known.

    We can see the point and provide more information about the SPAEF metric now. We have also shortened the section with the Mann-Kendall trend test.

Section 4.1: Taylor diagrams don't provide information about possible bias. I suggest adding this piece of information.

    Indeed, there were no bias information included. Therefore, we have added a table (Table 4) with bias values from the spatially and temporally averaged monthly values of the three

relevant SPEI variables along with some text for description: "Table 4 shows the bias values from the spatially and temporally averaged monthly values of the three variables compared to E-OBS. The highest spread among the models is found for the precipitation, which was expectable due to the higher variability of this variable. COSMO-CLM is the only RCM with a dry bias and also holds the lowest mean bias value (-1.5 mm), while RCA4 has by far the highest bias value (32 mm). The WRF@5 km bias value (16.1 mm) is almost twice as high as of its 15 km counterpart (8.3 mm). For $T_{max}$ the highest mean bias value is held by RCA4 (-1.8 °C), the lowest by REMO (0.2 °C). ALADIN, REMO and RegCM show a warm bias, while the other RCMs have a cold bias. Regarding $T_{min}$, RACMO is the RCM with the highest mean bias value (-1.7 °C), WRF@5 km the one with the lowest (-0.2 °C). COSMO-CLM, ALADIN, REMO and RegCM show warm bias, the other RCMs a cold bias.". Little text referring to this was also added to the conclusions: "Furthermore, there seem to be no correlation between the RCM bias values (Table 4) and the respective SPEI performances.".

L296: from 50, I guess.

Thanks for the hint. It should be "from 12.5 to 5 km". We have changed the sentence (also after a comment of reviewer 2) to "From our results, we obtain that, if existent, the benefits of a resolution increase from 12.5 to 5 km are less distinct.".

L314: I suggest removing Table 4 and introducing the mean correlation coefficient as an inset in the respective map (or near the title).

We agree. Table 4 was removed and the mean correlation coefficient values were added underneath the title of the respective models.

Section 4.3: why not use WRF@15 km here, at least in Table 5? It would further highlight the benefits of model settings.

We agree with the reviewer. We have added the map of the WRF@15 km run to Figure 3 and the relevant scores to Table 5. Figure 3 will then contain both WRF runs next to the E-OBS reference. Additional text was added in this section:

"The WRF@15 km domain shows more similarity with the E-OBS domain regarding the values, but the spatial distribution is different. This is underlined by the close SPEI-3 domain mean value (-1.81 compared to -1.90 of E-OBS), the almost exact area under drought (AUD) value (81.5 % compared to 81.7 %) and the lowest mean bias value (-0.08 SPEI units), but the low SPAEF value (0.10) in Table 5." and

"In fact, it is evident that the WRF@15 km run performs better in all scores except the SPAEF value (Table 5), which indicates the higher relevance of the model settings in this respect. This shows that, in some aspects, a lower resolution can also lead to better agreement with the reference compared to the higher resolution of the same model run.".

Some text referring to this was also added in the conclusions: "In terms of reproducing the drought event 2003, the model settings of WRF are determining for the highest agreement with the reference, since the 15 km run performs better than its 5 km counterpart. The event is not well captured by any of the other RCMs.".

Table 5: are the Mean SPEI values and the other statistics averaged over the German territory or the whole domain?

They were averaged over the whole domain.

L347: … of this RCM …

> Correct. *Of* was included: "The only exception is ALADIN with the maximum value of 0.55. REMO holds the lowest SPAEF value (-1.89), which completes the overall bad performance of this RCM in this regard."

L355: in the authors' opinion, why do we see these results with the Tiedtke scheme? However, from Table 2, I observe that also RegCM uses Tiedtke.

> We have further analyzed the relevant time steps (months) of the spatially averaged time series from the three meteorological variables to check if the deviations from the reference are also the highest in this case. It turned out that the RCMs which were run with the Tiedtke convection scheme do not show extraordinary bias values here. From this we conclude that the deterioration in SPEI-3 performance for the 2003 event is rather not related to the Tiedtke scheme. The fact that RegCM was also run with Tiedtke corroborates this conclusion. Consequently, we have removed the section in question.

Figs. 4 and 5: I would merge them. In general, I suggest always considering as two different configurations WRF@5km and WRF@15km (see my comment to Section 4.3).

> We agree that merging the two figures would make sense, but see the difficulty in the resulting figure dimensions since this would result in nine panels. As a compromise we would like to keep the Figs. 4 and 5 the way they are and add the WRF@15 km scores in the Tables 7–9 in the drought characteristics section. Now the WRF@15 km information is included in every single section. This implies also some adjustments in the results and conclusions section.

Section 4.5.1: E-OBS drought frequency looks too high for some areas (up to 22/24 times over 30 years). What is the reference period on which the index is calculated? 1980-2009?

> We can see that the drought frequency values appear relatively high for some areas. It must be kept in mind that already events with an SPEI-3 value equal to or below -1 are considered as droughts here. This does not necessarily imply drought events to be severe or extreme. Due to the definition of the SPEI, this can also imply a just drier than normal period (dry anomaly), which is then considered as a drought event. This can also happen in autumn and winter months. For clarification, some description for the proper classification was added to the text: "This may appear relatively high at first. It needs to be kept in mind that already events with an SPEI-3 value equal to or below -1 are considered as droughts (see Sect. 3.4), meaning that already moderate droughts (see Table 3) are taken into account. This does not necessarily imply drought events to be severe or extreme. Due to the definition of the SPEI, this can also imply just a drier than normal period, which is then considered as a drought event. This can also happen in any other season than summer.".
> Yes, the reference period for the index calculation is 1980–2009.

L414: so, it is mean absolute error (MAE).

> Correct, it is the mean absolute error. This information was added to the text: "For the drought characteristics we used the mean absolute error (MAE) as a measure for the domain mean bias (third column in Table 6) since values with opposite signs can balance each other out, thus making the information less meaningful."

Sections 4.4 and 4.5: discussion, specifically in terms of comparison to existing literature, is mostly missing.

We can see the point here. Searching for literature dealing with similar topics (trend detection and drought characteristics) for the same or similar region was not very fruitful. However, some discussion was added to section 4.4 regarding model capabilities to reproduce observed trends using the Mann-Kendall trend test: "Nasrollahi et al. (2015) applied the Mann-Kendall trend test to the outputs of 41 CMIP5 models to evaluate their ability to replicate observed drought trends on the global scale between 1901 – 2005. They used the SPI-3 as drought index (and SPI-6 in the supporting material). Their results revealed that about 75 % of the models reproduce the global drying trend, but most models fail at reproducing regional wetting and drying trends (at most about 40 % with agreement). In most locations, less than 10 % of the models showed agreement with the observations. Greater agreement was found in higher latitudes. Um et al. (2017) also performed the Mann-Kendall trend test on grid cell based SPEI-12 time series from outputs of four (HadGEM3-RA, MM5, RegCM4 and RSM) RCMs from CORDEX East Asia and of their ensemble mean for the time period 1980–2005 over East Asia. They found distinct differences among the single model outputs regarding their capability to reproduce observed drying and wetting trends. While HadGEM3-RA and MM5 generally captured the proper trends, RegCM4 and RSM were only partially successful. This is why the ensemble mean showed relatively poor performance compared to the two former RCMs. These results highlight the spread in the model's capability in reproducing observed trends of wetting and drying, which is found in this study as well.".

Regarding the drought characteristics, matching literature is sparse. Moreover, different methods, time and aggregation scales, definitions, reference data sets etc. make comparisons with existing literature not really meaningful in most cases.

Conclusions: maybe, the main achievements of the paper can be highlighted with bullet points (e.g., increased resolution and setup are not useful for drought characteristics; they are helpful for correlations; they are useful for trends, etc.).

We thank for the suggestion and comprehend that this would make it very clear. Nevertheless, we would like to keep the format in continuous text. Also, because it appears more common in NHESS.

**Response to RC2 on nhess-2022-162**

The manuscript describes an evaluation study of EURO-CORDEX hindcast simulations and additional WRF simulations with focus on the SPEI drought index. The methods are clearly described and mostly sound. In particular, the authors discuss the potential benefits of higher spatial resolution and regionally-tuned model setup, which may help to design efficient yet suitable model setups in future studies. The manuscript is suggested for publication after minor revision as described below.

Minor Revisions suggested:

Please provide your motivation for the model selection you applied (only 6 simulations). E.g, I'm pretty sure that EURO-11 simulations with several WRF configurations are available since many years. Why are they not used in this study?

> The models selected are all those that were available at that time that met two requirements: containing the necessary variables (precipitation, Tmax and Tmin) and coverage of the period 1980–2009. To make this clearer in the text, we have changed it in L131 to: "At the time of selection, these were all available model runs that cover the study period 1980–2009 and contain the relevant meteorological variables needed for the analysis."

In most practical applications, RCM simulations are only used after bias correction. Did you analyze the effect of bias correction (of temperature and precipitation) on your results? Please comment.

> Bias correction is usually based on observational data sets and especially meant for purposes of future projections, which are out of scope for this study. We aim at evaluating RCM performances regarding their capabilities to simulate droughts, not at bias correction (which could be the next step for a projections study). Here, the RCMs were forced by reanalysis data, where observations are already included. Since also requested by reviewer 1, to give the reader an idea of the biases involved, we have calculated the spatially and temporally averaged bias values of the monthly values of the relevant variables and have added them (Table 4) along with some description: "Table 4 shows the bias values from the spatially and temporally averaged monthly values of the three variables compared to E-OBS. The highest spread among the models is found for the precipitation, which was expectable due to the higher variability of this variable. COSMO-CLM is the only RCM with a dry bias and also holds the lowest mean bias value (-1.5 mm), while RCA4 has by far the highest bias value (32 mm). The WRF@5 km bias value (16.1 mm) is almost twice as high as of its 15 km counterpart (8.3 mm). For $T_{max}$ the highest mean bias value is held by RCA4 (-1.8 °C), the lowest by REMO (0.2 °C). ALADIN, REMO and RegCM show a warm bias, while the other RCMs have a cold bias. Regarding $T_{min}$, RACMO is the RCM with the highest mean bias value (-1.7 °C), WRF@5 km the one with the lowest (-0.2 °C). COSMO-CLM, ALADIN, REMO and RegCM show warm bias, the other RCMs a cold bias.". No relation could be detected between the bias values and the respective RCM SPEI performances. This suggests that a bias correction would probably not have much effect. This finding we have added to the conclusions as well: "Furthermore, there seem to be no correlation between the RCM bias values (Table 4) and the respective SPEI performances.".

I would like to see a critical discussion on the practical relevance of the SPEI in climate change (i.e. trend) studies. Since NHESS has a focus on hazards and the hazard is rather the agricultural drought than the meteorological drought, the question arises to which degree the SPEI is able to describe drought hazard and what the of the SPEI to serve as proxy for agricultural droughts are. E.g., SPEI does not cover the effect of increasing surface runoff during heavy precipitation events, which, however, is a loss in the soil moisture budget. Similarly, SPEI does not regard increased transpiration due to longer vegetation periods in a warmer climate. I respect that this is not the main topic of the manuscript and

therefore cannot be treated in a quantitative manner, but it is a very important boundary information and should be discussed in the conclusions and/or introduction sections.

> We can understand the points listed here. The SPEI describes the rainfall deficit in a certain time period based on a reference period. It does not describe any further impacts except that a lack of soil moisture (especially important for agriculture) or discharge, water levels etc. is assumed after a certain number of months with dry anomaly. As stated, the points mentioned are especially relevant in the climate change context, but climate change was not an issue in the strict sense of this study. Nevertheless, we see the importance of this point and have added some words to the methods section 3.2: "In this context it should be emphasized that the SPEI (and also SPI) has limitations regarding the practical relevance for climate change, when the focus is primarily on impacts. Apart from an implied lack of soil moisture (agricultural drought) and decline of streamflow, groundwater, reservoir and lake levels (hydrological drought), which completely rely on the degree of dry anomaly over a certain time period, impacts going beyond this are not addressed. Due to the complete reliance on dry anomaly, effects of a warming world (e.g., longer vegetation period and thus modified transpiration behavior), cannot be included either or would be considered only indirectly.".

L295: "Therefore, it can be assumed that if existent, the benefits of a resolution increase from 5 to 12.5 km are less distinct," A single simulation does not allow drawing such general conclusions. This statement refers also to other conclusions (e.g. line 321, 322).

> True, a single simulation is not enough for such clear and strict general conclusion. Thus, the sentence was adjusted to "From our results, we obtain that, if existent, the benefits of a resolution increase from 12.5 to 5 km are less distinct.". The sentence in line 321 will be adjusted to: "Our findings indicate that the WRF benefits can be attributed to the WRF model settings and not to the increased resolution.". Moreover, in the sections 4.3 ("Since WRF@5 km did not perform best in any of the categories in this case, there does not seem to be any benefit of increased model resolution and model settings in this regard in our results.") and 4.4 ("As seen, these benefits are not primarily due to increased resolution, but to the model settings, highlighting the high importance of model configurations tailored to the target region for our case.") some sentences were changed to make a clearer connection to our study.

L329: "Because of the results in the previous section, here we focus on the values from the WRF@5 km run in direct comparison to the reference values from OBS (Figure 3)." In the previous section, you argue that the 15km WRF simulation outperforms the 5km simulation, at least for temporal correlation. How does this lead to the decision to use only the 5km simulation in the subsequent section?

> This is a valid objection. Therefore, we have added the WRF@15 km map to Figure 3 and the relevant scores to Table 5. That implies adding some more text to the section:

> "The WRF@15 km domain shows more similarity with the E-OBS domain regarding the values, but the spatial distribution is different. This is underlined by the close SPEI-3 domain mean value (-1.81 compared to -1.90 of E-OBS), the almost exact area under drought (AUD) value (81.5 % compared to 81.7 %) and the lowest mean bias value (-0.08 SPEI units), but the low SPAEF value (0.10) in Table 5." and

> "In fact, it is evident that the WRF@15 km run performs better in all scores except the SPAEF value (Table 5), which indicates the higher relevance of the model settings in this respect. This shows that, in some aspects, a lower resolution can also lead to better agreement with the reference compared to the higher resolution of the same model run.".

> Little text referring to this was also added in the conclusions: "In terms of reproducing the drought event 2003, the model settings of WRF are determining for the highest agreement with the reference, since the 15 km run performs better than its 5 km counterpart. The event is not well captured by any of the other RCMs.".

More generally speaking: By (partly) removing the 15km WRF simulation from the analysis, you lose the option to directly compare it to the WRF 5km simulation and therefore you lose the most direct indicator for the effect of the spatial model resolution. On the other hand, major results of the study are statements like "computation resources could therefore be saved, since a coarser resolution can provide similar results" (Abstract) or "WRF's increased resolution and setup is turned out to be beneficial in the analysis of the monthly values of the meteorological variables and the correlations of the SPEI time series". Therefore, I suggest keeping both the 5km and 15km WRF simulations in each part of the analysis in order to better support your conclusions. (Side benefit: The additional Fig.5 could be avoided, if you had the 15km simulation in the general analysis (Figure 4).)

> We agree that more information of the 15 km simulation would be desirable. Therefore, we have added information from the WRF@15 km run in every section, at least in the tables and give further descriptions in the text, e.g., to compare the two WRF runs with each other. Adding one more domain map to the figures would increase the figure dimensions, since there would be nine domain maps in total per figure. Especially in section *4.5 Drought Characteristics Analysis* we consider the information from the tables as more meaningful. Thus, Figure 5 needs to be kept this way.